# The DANish Disseminated Intravascular Coagulation (DANDIC) Cohort Study: Time Trends in Incidence and Short-Term Mortality

**DOI:** 10.3390/jcm13195896

**Published:** 2024-10-02

**Authors:** Simon Flæng, Asger Granfeldt, Henrik Toft Sørensen, Kasper Adelborg

**Affiliations:** 1Department of Clinical Epidemiology, Aarhus University and Aarhus University Hospital, 8200 Aarhus, Denmark; 2Department of Clinical Medicine, Aarhus University, 8200 Aarhus, Denmark; 3Department of Anaesthesiology and Intensive Care, Aarhus University Hospital, 8200 Aarhus, Denmark; 4Department of Clinical Biochemistry, Gødstrup Hospital, 7400 Herning, Denmark

**Keywords:** disseminated intravascular coagulation, epidemiology, incidence, mortality

## Abstract

**Background:** Disseminated intravascular coagulation (DIC) is a severe condition affecting the coagulation system. However, current knowledge regarding its incidence and mortality remains limited. In this study, we examined the incidence and mortality of DIC, including time trends, in Denmark. **Methods:** In this population-based cohort study, potential DIC cases were identified through the hospital laboratory database in the Central Denmark Region which has a population of approximately 1.3 million residents. Eligibility criteria were age above 18 years, a positive DIC score, and a disease associated with DIC. All eligible patients underwent a review of their medical records. Follow-up started on the date of a patient’s first positive DIC score. Age- and sex-standardized incidence rates were calculated using year-specific DIC events as the numerator and the adult population of the Central Denmark Region as the denominator. All-cause 30-day mortality in the DIC cohort was computed based on Kaplan–Meier estimates and the mortality rates between subgroups were examined using logistic regression. **Results:** Among the 40,534 patients for whom all DIC biomarkers were measured on the same date, 6748 had a positive DIC score. Of these, 2565 were included in the cohort. The median age was 64 years, and 56.1% were men. The overall incidence rate per 100,000 person years declined during the study period, from 33.1 in 2013 to 24.0 in 2020. Thirty-day all-cause mortality was 35% in 2013 and 41.3% in 2020. **Conclusions:** The overall incidence rate of DIC declined between 2013 and 2020, mainly reflecting a declining incidence among patients with infection-associated DIC. Mortality did not improve.

## 1. Introduction

Disseminated intravascular coagulation (DIC), a serious condition of the coagulation system, is associated with high mortality and limited treatment options [1,2]. Although DIC is a devastating condition, few studies have examined time trends in terms of incidence and short-term mortality.

Singh et al. examined the incidence of DIC in patients admitted to the intensive care unit (ICU) in a single-centre, population-based, retrospective cohort study in the U.S. and observed a decrease in the overall incidence rate from 2004 to 2010 [3]. Furthermore, they found that in-hospital mortality remained unchanged during the study period [3]. Murata et al. and Yamakawa et al. identified patients with DIC in the Japanese Diagnosis Procedure Combination inpatient database [4,5]. Murata et al. reported a decrease in 28-day mortality for infection-associated DIC between 2010 and 2012 but observed no differences in patients with other diseases associated with DIC. Yamakawa et al. observed a decrease in 28-day mortality from 2010 to 2017. This decrease was particularly evident in patients with sepsis and leukaemia [4,5].

However, research to date investigating trends in incidence and mortality of DIC had several limitations. Singh et al.’s study was limited to patients admitted to the ICU, was conducted more than 10 years ago, and was limited in size. Murata et al. and Yamakawa et al. relied on ICD-10 diagnostic codes to identify patients with DIC, and the study by Murata et al. was conducted over a short period of time. Updated information on the occurrence and prognosis of DIC is thus needed. The aim of the current study was to examine time trends in DIC incidence in the general population and 30-day mortality rates among patients with DIC.

## 2. Methods and Materials

### 2.1. Design and Setting

This Danish population-based historical cohort study was conducted in the Central Denmark Region, which has approximately 1.3 million residents and covers 20% of the Danish population [6]. The cohort was drawn from all hospitals in the region, comprising six regional hospitals and one university hospital [7]. The study period initially spanned January 2011 to July 2021. However, a review of medical records indicated limited data availability, owing to delayed implementation of electronic medical records in 2011 and 2012. In addition, data were accessible for only the first half of 2021. Consequently, we changed the study period to the years 2013 to 2020.

Denmark provides free universal tax-supported healthcare with access to both general practitioners and hospitals [8]. All Danish residents receive a unique personal registration number at birth [9], which enables data linkage across various national registries on an individual level. Data on diseases associated with DIC and exclusion criteria were obtained from electronic medical records. Population data for the Central Denmark Region were acquired from Statistics Denmark. Data on levels of comorbidity, defined according to Charlson Comorbidity Index (CCI) scores, were derived from the Danish National Patient Registry, which covers all Danish hospitals (relevant ICD-10 discharge codes are provided in Appendix A). This registry was also used to determine the number of patients per year with a disease associated with DIC in the entire Danish population aged 20 years and older (relevant ICD-10 discharge codes are provided in Appendix A). Mortality data were obtained from the Danish Civil Registration System [9,10].

### 2.2. Study Cohort

A detailed description of the cohort has been published [7]. In brief, patients were identified from the Central Denmark Region’s hospital laboratory database, based on the biomarkers in the International Society on Thrombosis and Haemostasis (ISTH) and the modified Japanese Association for Acute Medicine (JAAM) DIC scoring systems [11,12]. Patients were eligible for inclusion if they were between the ages of 18 and 100 and had an ISTH DIC score ≥ 5 or a JAAM DIC score ≥ 4 (the ISTH and JAAM DIC scoring algorithms are available in Appendix A). All patients eligible for inclusion underwent review of their medical records at the time of their first positive DIC score. In addition to a positive DIC score, patients were required to have a disease associated with DIC. These included infection, malignancy, obstetric complications, cardiac arrest, trauma, and other conditions such as severe pancreatitis [4,13,14,15,16]. If a patient suffered from multiple diseases associated with DIC, the medical record reviewer assessed which disease contributed most to DIC development. This was registered as the primary disease associated with DIC and the remaining diseases were registered as secondary diseases associated with DIC (an overview of the overlap between the primary and secondary DIC associated diseases is presented in Table 1). The exclusion criteria consisted of other conditions affecting the coagulation system, e.g., treatment with vitamin K antagonists within a week before a positive DIC score. Patients with a known haematological malignancy and prevalent thrombocytopenia (platelet count < 120 × 10^9^/L), or no platelet count within three months of a DIC score consistent with DIC, were excluded. Also, patients with known liver disease and a prevalent prolonged international normalized ratio (INR) (>1.2) and/or thrombocytopenia (platelet count < 120 × 10^9^/L), or no measurement of these biomarkers consistent with DIC within three months of a DIC score, were excluded. When a patient was included in the cohort, follow-up started on the date of the first positive DIC score. To characterize patients, we obtained data on anticoagulant treatment, haemostatic treatment, and blood transfusions from the electronic medical records. Furthermore, we collected information on the location of infection, blood cultures, and presence of septic shock for patients with infection-associated DIC. Tumour histology was obtained for patients with malignancy-associated DIC, while the type of obstetric complication, parity, and gestational age in weeks were collected for patients with obstetric complications. Note that a patient could suffer from multiple infections, malignant disease, or obstetric complications simultaneously.

### 2.3. Statistical Analysis

Data are summarized as medians with 25th and 75th percentiles for continuous variables and counts and percentages for categorial data. Incidence rates were computed with year-specific DIC events as the numerator and the adult population of the Central Denmark Region, retrieved mid-year, as the denominator. For obstetric complications, the population at risk was limited to pregnant women in the Central Denmark Region. Because data on the prevalence of pregnancies were unavailable, the sum of births and abortions per year was used as a proxy for conceptions. Incidence rates were standardized by age and sex, with the 2020 background population serving as reference, and presented as incidence rates per 100,000 person years with corresponding 95% confidence intervals (95% CI). Subgroup analyses were performed by age, sex, baseline DIC scores, and primary disease associated with DIC. The age groups were 18–39 years, 40–59 years, 60–79 years, and 80–99 years. Baseline DIC scores were divided into subgroups as follows: ISTH scores of 5–6 and JAAM scores of 4–6 were defined as low, and ISTH and JAAM scores of 7–8 were defined as high.

Subgroup comparisons of general population incidence by age and sex were examined using a Poisson regression model and presented as incidence rate ratios (IRRs), both crude and adjusted by age group and sex.

Kaplan–Meier estimates were used to compute 30-day all-cause mortality for patients included in the DIC cohort. Analysis was performed overall and in subgroups defined by age, sex, baseline DIC score, and primary disease associated with DIC. Logistic regression analyses were used for subgroup comparisons of 30-day mortality by age, sex, and baseline DIC score, with results presented as odds ratios (ORs), both crude and adjusted by age group, sex, and level of comorbidity according to CCI scores. Malignancy is a disease associated with DIC and a comorbidity included in the CCI. Malignancy was included in the calculation of the CCI for all patients.

Time trends in incidence and mortality were evaluated using Joinpoint regression and presented as the annual percentage change (APC) if no Joinpoints were present or as the average annual percentage change (AAPC) if one or more Joinpoint was present [17]. The number of Joinpoints was selected using the Weighted Bayesian Information Criterion (WBIC) and confidence intervals were determined using the Empirical Quantile method. Smoothed lines depicting trends over time were created with locally weighted smoothing (LOESS).

Data analysis was performed in Rstudio v. 4.3.0, and Joinpoint analysis was performed in Joinpoint v 5.0.2. Data obtained from medical records were stored in REDCap.

### 2.4. Approvals

The project was approved by the Danish Patient Safety Authority (31-1521-452), the Central Denmark Region (1-45-70-83-21), the Danish Data Protection Agency (1-16-02-258-21), and all the hospital chairs. Register-based studies require no ethical approval in Denmark.

## 3. Results

From January 2011 to June 2021, a total of 40,534 patients had same-day measurements of all DIC biomarkers included in either the ISTH or JAAM scoring system (the number of patients with all biomarkers measured on the same date and the number of positive and negative DIC scores per year are illustrated in Appendix A). Of these patients, 3546 (8.7%) had an ISTH DIC score of 5 or higher, and 5839 (14.4%) had a JAAM DIC score of 4 or higher, yielding 6748 unique patients with a positive DIC score. After the medical record review of all potential DIC cases, 1575 patients were excluded because of limited data availability in 2011, 2012, and 2021, and 2608 were excluded because one or more exclusion criteria were met. The final analytic cohort consisted of 2565 patients with DIC from 2013 to 2020 (Figure 1). The most common disease associated with DIC was infection, followed by cardiac arrest, obstetric complications, and trauma. The median age was 64 years, and 56.1% of patients were men (Table 1).

### 3.1. Incidence over Time

The overall incidence rate per 100,000 person years, standardized by age and sex, declined during the observation period, from 33.1 in 2013 to 24.0 in 2020 (AAPC: −5.4% [95% CI, −8.3% to −2.9%]). The incidence rate in the 40–59-year age group declined from 21.5 in 2013 to 16.5 in 2020 (AAPC: −5.2% [95% CI, −9.6% to −0.3%]), and that in the 60–79-year age group declined from 61.8 in 2013 to 40.9 in 2020 (AAPC: −7.4% [95% CI, −11.0% to −4.5%]). In contrast, the incidence rates did not change over time in the 18–39- and 80–99-year age groups. Both men and women showed a decline in DIC incidence over time. The incidence rate declined from 33.6 in 2013 to 28.6 in 2020 among men (AAPC: −3.9% [95% CI, −6.9% to −1.5%]) and from 28.5 in 2013 to 19.6 in 2020 among women (AAPC: −5.9% [95% CI, −7.4% to −4.6%]). The age- and sex-standardized incidence rates of patients with low baseline DIC scores also declined over time. DIC incidence decreased from 13.2 in 2013 to 10.8 in 2020 among patients with low baseline ISTH DIC scores (AAPC: −4.7% [95% CI, −10.3% to −0.3%]) and from 24.2 in 2013 to 19.4 in 2020 among patients with low baseline JAAM DIC scores (AAPC: −3.9% [95% CI, −6.2% to −1.6%]). The incidence rate for patients with high baseline DIC scores remained stable during the observation period (Figure 2).

The incidence rate of infection-associated DIC declined from 22.6 in 2013 to 15.9 in 2020 (AAPC: −6.8% [95% CI, −10.0% to −3.7%]). Patients with malignancy-associated DIC showed an increase in the DIC incidence rate from 0.2 in 2013 to 0.6 in 2013 (AAPC: 17.4% [95% CI, 2.1% to 45.8%]). The incidence of DIC associated with obstetric complications initially increased from 160.0 in 2013 to 231.8 in 2016 (APC: 12.1% [95% CI, 2.1% to 33.4%]), then declined to 127.5 in 2020 (APC −15.3% [95% CI, −25.2% to −9.5%]. However, the declining DIC incidence from 2016 onward corresponded to a general decline in the number of patients with obstetric complications in the entire Danish population aged 20 years and older (Appendix A). Overall, the incidence of DIC associated with obstetric complications declined throughout the study period (AAPC: −4.5% [95% CI, −8.8% to 0.0%]). No temporal changes in incidence were observed among patients with cardiac arrect, trauma, or other diseases associated with DIC (Figure 3).

### 3.2. DIC Incidence in Subgroups

The incidence of DIC increased with age. Using the age group from 18 to 39 as reference and adjusting for sex, the 40–59-year age group had a 1.41 (95% CI, 1.25–1.59) times higher incidence rate of DIC; the 60–79-year age group had a 3.49 (95% CI, 3.99–5.32) times higher incidence rate of DIC; and the 80–99-year age group had a 4.61 (95% CI, 3.99–5.32) times higher incidence rate of DIC. Men had a 1.35 (95% CI, 1.25–1.46) times higher incidence of DIC than women after adjusting for age (Table 2).

### 3.3. Mortality over Time

The overall 30-day all-cause mortalityof patients in the cohort did not improve during the study period. It was 35% in 2013 and 41.3% in 2020 (AAPC: 1.0% [95% CI, −1.8% to 3.5%]). Thirty-day mortality increased in the youngest group, from 4.1% in 2013 to 12.2% in 2020 (AAPC: 13.5% [95% CI, 0.2% to 34.9%]), while no changes were observed in the other age groups. Moreover, no changes in 30-day mortality were observed in the subgroups by sex or baseline DIC score (Figure 4).

Mortality in patients with trauma-associated DIC increased from 8.7% in 2013 to 33.3% in 2020 (AAPC: 19.1% [95% CI, 5.3% to 43.0%]), while the remaining diseases associated with DIC displayed no temporal changes. Malignancy as the primary disease associated with DIC was present in too few cases to permit analysis and therefore was excluded from the survival analysis (Figure 5).

### 3.4. Mortality in Subgroups

The 30-day mortality rate increased with age. When using the youngest group from 18 to 39 years of age as a reference, the adjusted OR was 2.89 (95% CI, 2.09–4.03) for patients 40–59 years of age; 4.35 (95% CI, 3.20–5.98) for patients 60–79 years of age; and 5.41 (95% CI, 4.49–9.20) for patients 80–99 years of age. Men had increased 30-day mortality compared to women with an OR of 1.34 (95% CI, 1.15–1.58), but after adjusting for age and comorbidities the OR decreased to 1.03 (95% CI, 0.87–1.22). Patients with a baseline ISTH score of 7–8 had an adjusted OR of 3.05 (95% CI, 2.08–4.53) compared to patients with a baseline ISTH score of 5–6. Patients with a baseline JAAM score of 7–8 had an adjusted OR of 1.65 (95% CI, 1.27–2.14) compared to patients with a baseline JAAM score of 4–6 (Table 3).

## 4. Discussion

### 4.1. Main Findings

The overall incidence of DIC generally declined throughout the study period. A decline in incidence over time was observed for both men and women, while patients with low baseline DIC scores and patients with high baseline scores had a stable incidence rate. Infection was the most common disease associated with DIC. The incidence of infection-associated DIC declined. As the majority of patients in the cohort had infection as the primary disease associated with DIC, the changes in the overall incidence of DIC mainly reflected the decreasing incidence in this subgroup. Furthermore, incidence increased with age and men had higher incidence than women.

The overall 30-day mortalityamong DIC patients did not improve during the observation period. Mortality increased with age, but no difference in mortality was observed between men and women. Patients with high baseline DIC scores had a higher mortality rate than patients with low baseline DIC scores.

### 4.2. Perspectives

Our findings of a declining overall incidence rate of DIC align with those of Singh et al., who reported a decline in the overall incidence rate of DIC from 26.2 in 2004 to 18.6 in 2010 [3]. Singh et al. attributed this decline to the substantial decrease in DIC among men in their study population. However, we observed a declining incidence in DIC among both men and women. Our cohort differs from that of Singh et al. in that we had a larger proportion of patients with DIC associated with obstetric complications. We observed declining DIC incidence in this subgroup, which consisted solely of women and might have contributed to the decline in incidence generally observed among women in our study. The decline in overall DIC incidence that we observed may be attributed primarily to the decline in infection-associated DIC, which accounted for most DIC cases. The increasing incidence among patients with malignancy-associated DIC had little impact on the overall incidence, as the affected group was small. One could argue that the decreasing incidence may be explained by improvements in healthcare delivery, such as better management of sepsis, which potentially could reduce the incidence of infection-associated DIC [18]. Singh et al. also reported increasing incidence with age and a higher incidence among men than women, corresponding to our findings.

Our observation of a lack of improvement in the 30-day mortality rate is consistent with the findings of Singh et al. but differs from the findings in the two Japanese studies based on data from the Japanese Diagnosis Procedure Combination inpatient database. Murata et al. reported a decrease in 28-day in-hospital mortality among patients with infection-associated DIC from 31.1% in 2010 to 27.7% in 2012, whereas Yamakawa et al. reported an overall decrease in 28-day mortality from 41.8% in 2010 and 36.1% in 2018 [4,5]. Murata et al. suggested that the decreased mortality is associated with new guidelines for DIC diagnosis and treatment in Japan, as well as advancements in treatment of infection-associated DIC with the use of recombinant thrombomodulin. Yamakawa et al. reported increased use of recombinant thrombomodulin throughout the study period and attributed the decreased mortality both to changes in anticoagulant treatment in patients with DIC and to important advances in treatment of the diseases associated with DIC, particularly sepsis and leukaemia. The difference in mortality rates between our study and the two Japanese studies might have been influenced by the different approaches to DIC-specific treatment [19]. Treatment with antithrombin substitution and recombinant thrombomodulin are recommended for sepsis-associated DIC in Japan but are not common in Danish clinical practice [1,20,21,22,23]. Furthermore, we excluded patients with haematological malignancies who had prevalent thrombocytopenia. Consequently, improvements in the treatment and prognosis of such diseases as leukaemia would have had smaller effects on the mortality observed in our study.

The lack of improvement in 30-day mortality could also be attributed to the declining incidence of low baseline DIC scores in our study. Patients with low baseline DIC scores had a lower mortality rate than patients with high baseline DIC scores; thus, the more severe cases with higher mortality persisted.

### 4.3. Strengths and Limitations

This study has several strengths. First, patient inclusion was based on a combination of laboratory data and medical record review rather than ICD-10 codes [24]. This supported a higher sensitivity for the diagnosis of DIC and more accurate DIC diagnoses, including information on patients’ baseline DIC scores. Second, data for the cohort covered all hospitals in the Central Denmark Region rather than being limited to a single hospital. Previous research has shown that patient characteristics and healthcare utilization are relatively homogeneous across Danish regions [6]. Third, all adult patients with measured DIC biomarkers were eligible for inclusion. The cohort thus was not restricted to patients with an ICU admission. Finally, the unique Danish personal registration number allows virtually complete follow-up, and the country’s free universal tax-supported healthcare system limits referral bias.

Several study limitations also should be noted. The process of electronic medical record review is susceptible to interobserver variability. To address this, authors KAD and SF reviewed and discussed the first 50 electronic medical records [7]. Also, left-truncation could be a potential source of bias [25]. Patients were assessed for inclusion at the first occurrence of a positive DIC score during the study period, but information on the presence of a positive DIC score before 2011 were not available. However, since we have biomarkers from 2011 to 2012 and we modified the study period to begin in 2013 due to the limited availability of electronic medical records, we know that none of the included patients had a positive DIC score in 2011 or 2012. Furthermore, the distinction between primary and secondary diseases associated with DIC was based on a clinical assessment by an author, who reviewed the medical records. DIC is multifactorial, and determining which disease contributed the most to the coagulopathy is a difficult task. This complexity could potentially introduce misclassification. However, this potential bias would only affect the subgroup analysis by diseases associated with DIC and not the overall findings. Another concern is that the mortality data were based on all-cause mortality and not DIC-specific mortality. Because we examined only 30-day mortality, the death of a patient was considered likely to be associated with DIC episodes. In addition, some patients might have had DIC but remained undiagnosed because the biomarkers were not measured, e.g., patients with end-stage malignant disease may have received only palliative treatment or the patient could have died before the measurement of the biomarkers. This might have led to underestimation of the incidence and mortality rates. Finally, comorbidity was assessed with CCI scores, which were calculated using ICD-10 hospital discharge codes from the Danish National Patient Registry. Therefore, a given comorbidity was included in our analysis only if it was registered at some point during a hospitalization. This requirement could lead to underestimation of comorbidities. Data on the number of patients with a disease associated with DIC in the entire Danish population were only available for patients over the age of 20.

## 5. Conclusions

The overall DIC incidence rate in the general population declined during the study period, largely attributed to declining incidence in patients with infection-associated DIC, who constituted most of the study population. The declining incidence could be due to better management of the diseases associated with DIC, particularly sepsis. At the same time, mortality did not improve, which could be influenced by a lack of DIC-specific treatment.

## Figures and Tables

**Figure 1 jcm-13-05896-f001:**
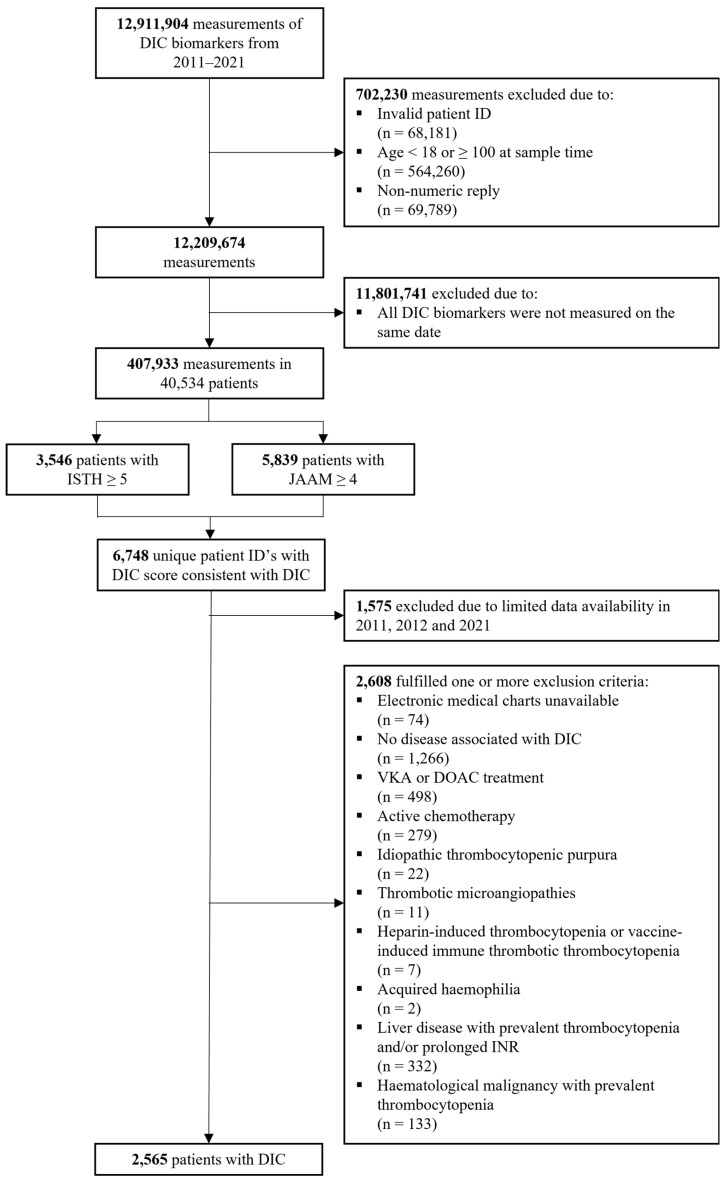
Flowchart of inclusion and exclusion. Abbreviations: ISTH: International Society on Thrombosis and Haemostasis; JAAM: Japanese Association of Acute Medicine; VKA: vitamin K antagonist; DOAC: direct oral anticoagulant; INR: international normalized ratio.

**Figure 2 jcm-13-05896-f002:**
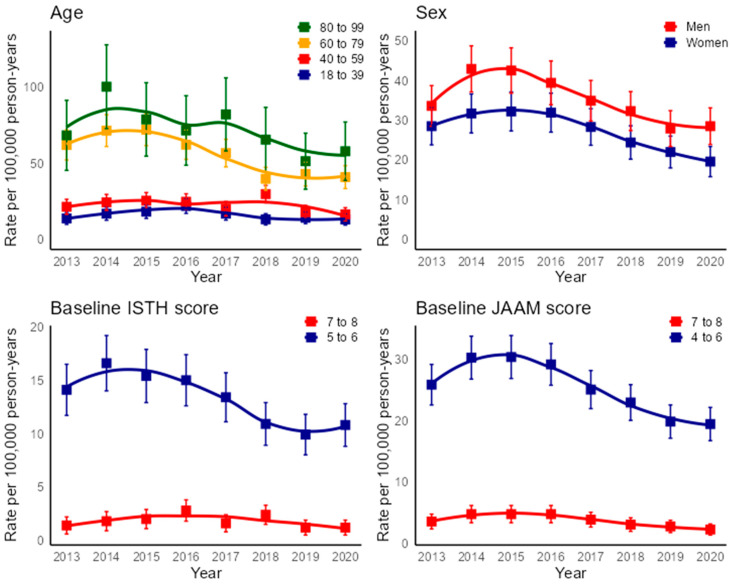
Incidence rate of DIC per 100,000 person years by age group, sex, and baseline DIC score. The 95% confidence intervals are represented by error bars, and the smoothed lines were created using the Loess technique. Abbreviations: ISTH: International Society on Thrombosis and Haemostasis; JAAM: Japanese Association of Acute Medicine.

**Figure 3 jcm-13-05896-f003:**
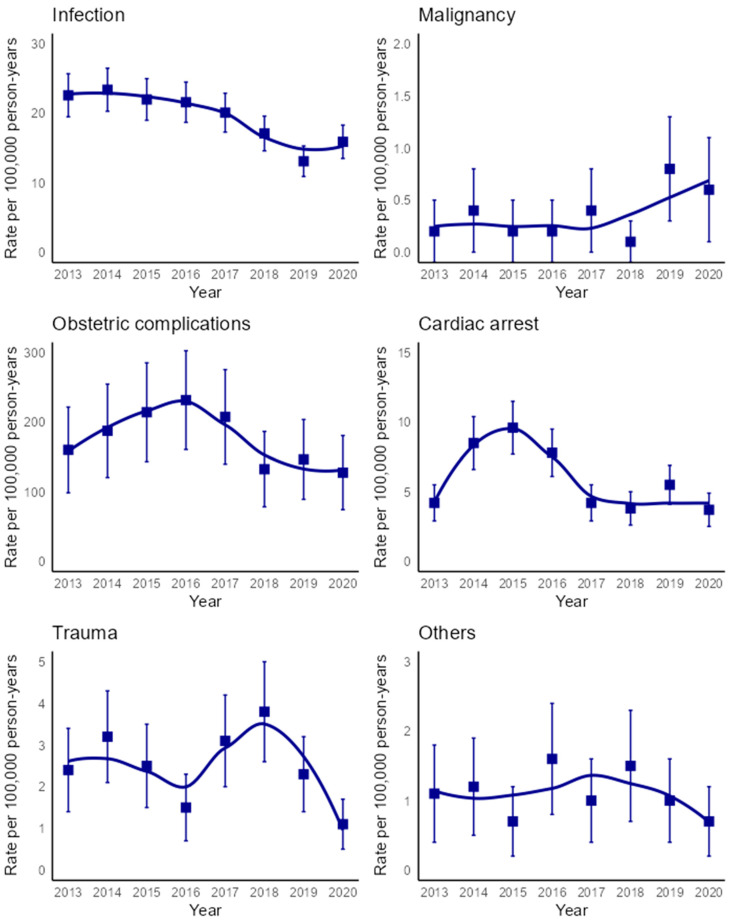
Incidence rate of DIC per 100,000 person years by primary disease associated with DIC. The 95% confidence intervals are represented by error bars, and the smoothed lines were created using the Loess technique.

**Figure 4 jcm-13-05896-f004:**
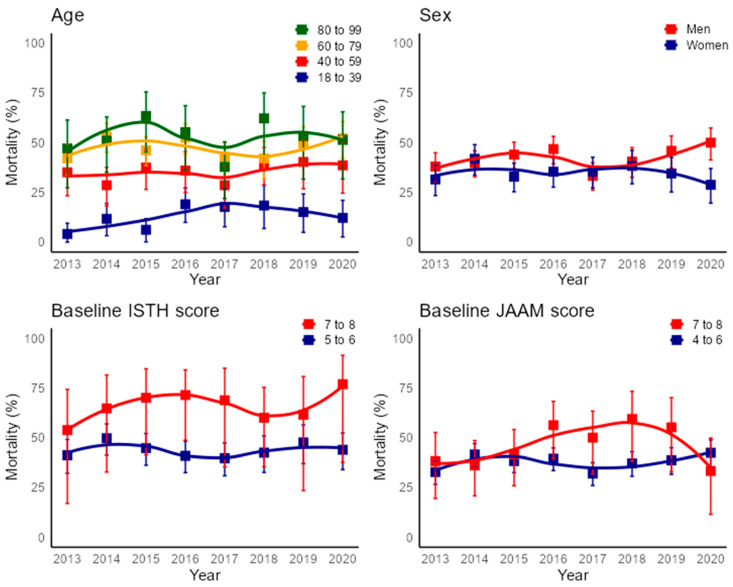
All-cause 30-day mortality time trends by age, sex, and baseline DIC score. The 95% confidence intervals are represented by error bars, and the smoothed lines were created using the Loess technique. Abbreviations: ISTH: International Society on Thrombosis and Haemostasis; JAAM: Japanese Association of Acute Medicine.

**Figure 5 jcm-13-05896-f005:**
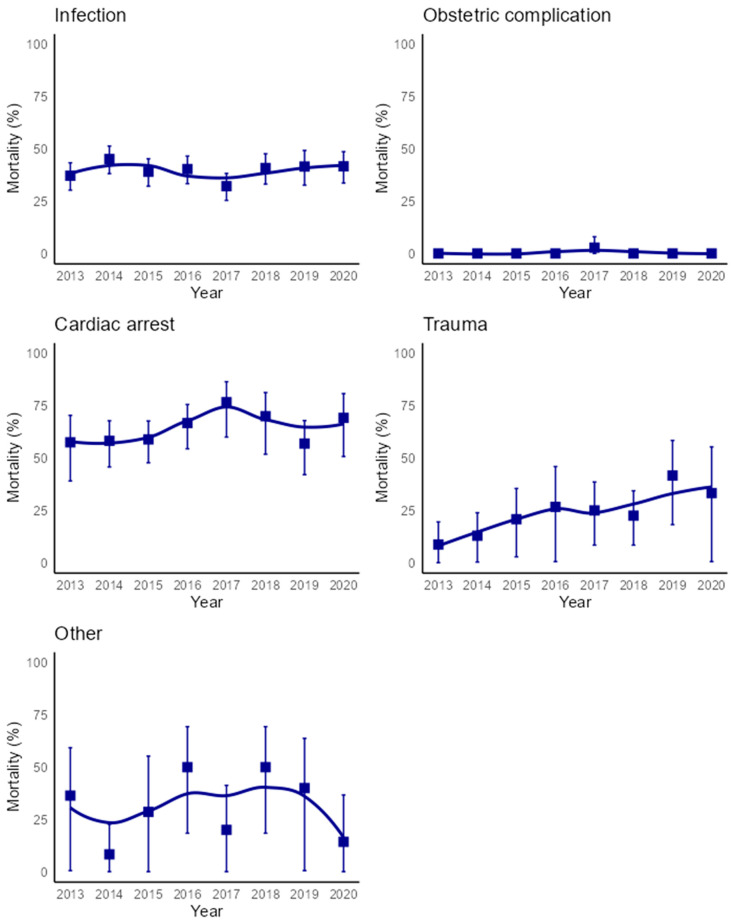
All-cause 30-day mortality by primary disease associated with DIC. The 95% confidence intervals are represented by error bars, and the smoothed lines were created using the Loess technique.

**Table 1 jcm-13-05896-t001:** Baseline characteristics of the study population, Central Denmark Region, Denmark, 2013–2020.

		Primary Disease Associated with DIC					
	Total	Infection	Malignancy	Obstetric Complications	Cardiac Arrest	Trauma	Others
Patients with DIC, n	2565	1537	28	238	472	201	89
Patients per year, n (%)							
2013	309 (12.0)	207 (13.5)	<5	26 (10.9)	40 (8.5)	23 (11.4)	11 (12.4)
2014	374 (14.6)	219 (14.2)	<5	30 (12.6)	79 (16.7)	31 (15.3)	12 (13.5)
2015	378 (14.7)	215 (14.0)	<5	35 (14.7)	95 (20.1)	24 (11.9)	7 (7.9)
2016	365 (14.2)	213 (13.9)	<5	41 (17.2)	78 (16.5)	15 (7.5)	16 (18.0)
2017	327 (12.7)	202 (13.1)	<5	36 (15.1)	43 (9.1)	32 (15.9)	10 (11.2)
2018	296 (11.5)	176 (11.5)	<5	23 (9.7)	40 (8.5)	40 (19.9)	16 (18.0)
2019	262 (10.2)	137 (8.9)	8 (28.6)	25 (10.5)	58 (12.3)	24 (11.9)	10 (11.2)
2020	254 (9.9)	168 (10.9)	6 (21.4)	22 (9.2)	39 (8.3)	12 (6.0)	7 (7.9)
Age, median (25th–75th percentile)	64 (48–74)	68 (57–76)	75 (65–81)	30 (27–34)	64 (51–72)	55 (37–71)	55 (43–71)
Age group, n (%)							
18–39	466 (18.2)	123 (8.0)	<5	222 (93.3)	47 (10.0)	56 (27.9)	17 (19.1)
40–59	624 (24.3)	353 (23.0)	<5	16 (6.7)	158 (33.5)	60 (29.9)	34 (38.2)
60–79	1164 (45.4)	831 (54.1)	13 (46.4)	<5	223 (47.2)	66 (32.8)	31 (34.8)
80–99	311 (12.1)	230 (15.0)	11 (39.3)	<5	44 (9.3)	19 (9.5)	7 (7.9)
Sex, n (%)							
Female	1126 (43.9)	643 (41.8)	11 (39.3)	238 (100.0)	147 (31.1)	50 (24.9)	37 (41.6)
Male	1439 (56.1)	894 (58.2)	17 (60.7)	–	325 (68.9)	151 (75.1)	52 (58.4)
Baseline ISTH score, n (%)							
5–6	1055 (41.1)	749 (48.7)	12 (42.9)	54 (22.7)	168 (35.6)	36 (17.9)	36 (40.4)
7–8	145 (5.7)	90 (5.9)	<5	<5	41 (8.7)	<5	7 (7.9)
Baseline JAAM score, n (%)							
4–6	2014 (78.5)	1161 (75.5)	18 (64.3)	203 (85.3)	381 (80.7)	181 (90.0)	70 (78.7)
7–8	302 (11.8)	189 (12.3)	<5	32 (13.4)	57 (12.1)	14 (7.0)	7 (7.9)
Secondary disease associated with DIC, n (%)							
Infection	126 (4.9)	–	7 (25.0)	<5	73 (15.5)	18 (9)	22 (24.7)
Malignancy	193 (7.5)	171 (11.1)	–	<5	<5	<5	<5
Obstetric complications	<5	<5	<5	–	<5	<5	<5
Cardiac arrest	108 (4.2)	<5	<5	<5	–	26 (12.9)	<5
Trauma	15 (0.6)	<5	<5	<5	6 (1.3)	–	<5
Others	27 (1.1)	<5	<5	<5	<5	<5	–
Charlson Comorbidity Index score, n (%)							
0	1207 (47.1)	582 (37.9)	<5	232 (97.5)	183 (38.8)	150 (74.6)	57 (64.0)
1–2	874 (34.1)	573 (37.3)	9 (32.1)	6 (2.5)	230 (48.7)	33 (16.4)	23 (25.8)
>2	484 (18.9)	382 (24.9)	16 (57.1)	<5	59 (12.5)	18 (9.0)	9 (10.1)
Most common comorbidity *	1. Cancer 2. Pulmonary disease 3. Diabetes	1. Cancer 2. Pulmonary disease 3. Diabetes	1. Cancer 2. Metastatic cancer 3. Pulmonary disease	1. Pulmonary disease 2. Renal disease/ diabetes/ cancer 3. –	1. Acute myocardial infarction 2. Congestive heart failure 3. Diabetes	1. Cancer 2. Cerebral vascular accident 3. Pulmonary disease	1. Peripheral vascular disease 2. Cancer/ Pulmonary disease/diabetes 3. –
Anticoagulant treatment, n (%) **							
Prophylactic LMWH ***	739 (28.8)	468 (30.4)	5 (17.9)	58 (24.4)	98 (20.8)	78 (38.8)	32 (36.0)
Therapeutic LMWH ***	207 (8.1)	145 (9.4)	<5	10 (4.2)	34 (7.2)	8 (4.0)	6 (6.7)
Unfractionated heparin	192 (7.5)	74 (4.8)	<5	<5	112 (23.7)	<5	<5
Antithrombin	169 (6.6)	78 (5.1)	<5	<5	87 (18.4)	<5	<5
Direct thrombin inhibitor	6 (0.2)	<5	<5	<5	<5	<5	<5
Haemostatic treatment, n (%) **							
Tranexamic acid	176 (6.9)	51 (3.3)	<5	61 (25.6)	15 (3.2)	45 (22.4)	<5
Protamine sulphate	56 (2.2)	26 (1.7)	<5	<5	29 (6.1)	<5	<5
Prothrombin complex concentrate	21 (0.8)	10 (0.7)	<5	<5	8 (1.7)	<5	<5
Fibrinogen concentrate	88 (3.4)	27 (1.8)	<5	6 (2.5)	34 (7.2)	17 (8.5)	<5
Phytomenadione	242 (9.4)	185 (12.0)	<5	5 (2.1)	20 (4.2)	10 (5.0)	18 (20.2)
Blood transfusions, n (%) **							
1–2 units	427 (16.6)	282 (18.3)	6 (21.4)	21 (8.8)	60 (12.7)	45 (22.4)	13 (14.6)
>2 units	893 (34.8)	510 (33.2)	8 (28.6)	31 (13.0)	217 (46.0)	102 (50.7)	25 (28.1)
Location of infection, n (%)							
Pulmonary		583 (37.9)					
Urogenital		217 (14.1)					
Gastrointestinal		425 (27.7)					
Skin or soft tissue		63 (4.1)					
Bone		7 (0.5)					
Catheter or other instruments		11 (0.7)					
Central nervous system		33 (2.1)					
Endocarditis		62 (4.0)					
Other		21 (1.4)					
Unknown		189 (12.3)					
Multiple sites of infection, n (%)		68 (4.4)					
Positive blood culturen (%)		613 (39.9)					
Septic shock, n (%)		570 (37.1)					
Tumour histology, n (%)							
Adenocarcinoma			12 (42.9)				
Uroepithelium			<5				
Haematological			7 (25)				
Other			<5				
Unknown			6 (21.4)				
Type of obstetric complication, n (%)							
Miscarriage				<5			
Preterm labour				6 (2.5)			
PROM				16 (6.7)			
IUGR				22 (9.2)			
Stillbirth				<5			
Preeclampsia				176 (73.9)			
HELLP syndrome				137 (57.6)			
Chorioamnionitis				<5			
Placental abruption				8 (3.4)			
Acute fatty liver				6 (2.5)			
Amniotic fluid embolism				<5			
Parity no., n (%)							
0				179 (75.2)			
1				39 (16.4)			
2				14 (5.9)			
>2				6 (2.5)			
Gestational week, median (25th–75th percentile)				37 (34–38)			
Other diseases associated with DIC, n (%)							
Reactions to toxins							5 (5.6)
Immunologic disorders							10 (11.2)
Organ damage							87 (97.8)
Heat stroke							<5

* According to the Charlson Comorbidity Index; ** Within one week before or after DIC diagnosis; *** The distinction between prophylactic and therapeutic treatments was based on the administered dose. Abbreviations: DIC: disseminated intravascular coagulation; HELLP: haemolysis, elevated liver enzymes and low platelets; IUGR: intrauterine growth restriction; ISTH: International Society on Thrombosis and Haemostasis; JAAM: Japanese Association of Acute Medicine; LMWH: low-molecular-weight heparin; PROM: premature rupture of membranes.

**Table 2 jcm-13-05896-t002:** Crude and adjusted incidence rate ratios by age, sex, and baseline DIC score.

	Incidence Rate Ratio (95% CI)	
Subgroup	Crude	Adjusted *
By age		
18–39 years	Reference group	Reference group
40–59 years	1.41 (1.25–1.59)	1.41 (1.25–1.59)
60–79 years	3.47 (3.11–3.86)	3.49 (3.14–3.89)
80–99 years	4.45 (3.85–5.13)	4.61 (3.99–5.32)
By sex		
Female	Reference group	Reference group
Male	1.29 (1.19–1.40)	1.35 (1.25–1.46)

* Adjusted for age group and sex.

**Table 3 jcm-13-05896-t003:** Comparison of 30-day all-cause mortality risk in subgroups.

			Odds Ratio (95% CI)	
Subgroup	At Risk	Events (%)	Crude	Adjusted *
By age				
18–39 years	465	62 (13%)	Reference group	Reference group
40–59 years	624	221 (35%)	3.57 (2.63–4.92)	2.89 (2.09–4.03)
60–79 years	1164	555 (48%)	5.96 (4.49–8.03)	4.35 (3.20–5.98)
80–99 years	311	163 (52%)	7.27 (5.16–10.35)	5.41 (3.78–7.82)
By sex				
Female	1126	395 (35%)	Reference group	Reference group
Male	1438	606 (42%)	1.34 (1.15–1.58)	1.03 (0.87–1.22)
By baseline ISTH score				
ISTH 5–6	1055	462 (44%)	Reference group	Reference group
ISTH 7–8	145	96 (66%)	2.51 (1.75–3.63)	3.05 (2.08–4.53)
By baseline JAAM score				
JAAM 4–6	2013	761 (38%)	Reference group	Reference group
JAAM 7–8	302	140 (46%)	1.42 (1.11–1.81)	1.65 (1.27–2.14)

* Adjusted for age group, sex, and comorbidity. Abbreviations: ISTH: International Society on Thrombosis and Haemostasis; JAAM: Japanese Association of Acute Medicine.

## Data Availability

The original contributions presented in the study are included in the article/Appendix A and further inquiries can be directed to the corresponding author.

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
