# Peer review of "The DANish Disseminated Intravascular Coagulation (DANDIC) Cohort Study: Time Trends in Incidence and Short-Term Mortality"

_jcm, 2024, doi:10.3390/jcm13195896_

Round 1

Reviewer 1 Report

Comments and Suggestions for Authors

The topic is interesting and important for acute medicine practitioners and hematologists. However few points need adjustments to improve the work. please address them carefully.

Line 76: in You should mention in the study design that it was a retrospective one.

Line 79: Why did you choose 20 years? adulthood is from 18!

Line 88: in Line 79 you mentioned the age 20 and more. Also in figure 1 you mentioned that you excluded patients older than 100 years, so mention that in the exclusion criteria.

Line 102: Why did you use this platelet level? thrombocytopenia is defined as platelet count less than 150 thousands.

Reviewer 2 Report

Comments and Suggestions for Authors

Authors examined time trends in DIC incidence in the general population and 30-day mortality rates among patients with DIC.

Although this manuscript is potentially interesting, several issues arise.

Cut off value of fibrin related markers is required.

SIRS score in JAAM score is required.

Tranexamic acid may be contraindication for septic DIC.

Table 1: There was no data for therapeutic LMWH.

Were there hematological malignancies?

Bleeding or organ failure symptom may be helpful.

The frequency of solid cancer was low.

Did this database include COVID-19?

Almost DIC patients were not treated with specific therapy, suggesting that the mortality of DIC patients without specific treatment was about 35-43%.

Reviewer 3 Report

Comments and Suggestions for Authors

based on your findings , what is the most appropriate score to use: ISTH or JAAM.

No additional remarks

Reviewer 4 Report

Comments and Suggestions for Authors

Dear Editor
I congratulate you on your achievement in the treatment of such a severe complication as DIC. However, analysing the publication raises questions that need to be answered.
From the tables presented, we learn that the number of infections complicated by coagulopathy decreased significantly in the period described ( from 207 to 137, 168 cases).
This decrease is due to improved medical standards.
Is it not an overinterpretation to aggregate the statistically significant number of patients with infections and DIC with other statistically significantly smaller groups (several and dozens of patients) and aggregate the conclusions.
The persistence of a similar number of deaths over the course of the follow-up confirms that the observed decrease in cases of patients with DIC applies only to severely ill patients.
In this sense, is it really the case that the therapy of DIC did not improve in the observation period, but only the therapy and prevention of infections, the number of which decreased significantly?

Round 2

Reviewer 2 Report

Comments and Suggestions for Authors

References are a biased destribution for critical care field.

Several papers in general medicine should be cyted. 

Reviewer 4 Report

Comments and Suggestions for Authors

The authors' corrections do not change the message of the paper. Combining statistically small groups with a large cohort and drawing general conclusions is a fault. The publication only shows the fact of a decrease in sepsis cases complicated by DIC. The data presented cannot be used to draw general conclusions about the whole problem of DIC. Even with the reference to the decrease in infections cited in the correction. The paper is methodologically flawed.

Comments on the Quality of English Language

Minor editing of English language
